# Self-Regulation Intervention Impact on Turkish Children with Emotional and Behavioral Disorder Risks

**DOI:** 10.3390/bs15040508

**Published:** 2025-04-10

**Authors:** Gamze Apaydın, Çığıl Aykut

**Affiliations:** Department of Special Education, Gazi University, 06560 Ankara, Türkiye; cigil@gazi.edu.tr

**Keywords:** self-regulation, emotional and behavioral disorders, at risk, preschool children, intervention

## Abstract

This study examined the effect of a self-regulation intervention package prepared for preschool children at risk for emotional and behavioral disorders (EBDs) in Türkiye on children’s self-regulation skills, social skills, problem behaviors, relationships with teachers, and peer acceptance. Ten children, five at risk of EBD and five with typical development (TD), participated in the study. This study used a single-group pretest-posttest design. The intervention lasted for eight weeks, two days a week. Follow-up data were collected three weeks after the intervention. It was found that the intervention package was effective in self-regulation, problem behavior, social skills, and peer acceptance variables of children at risk of EBD (z = −2.02, *p* < 0.05, r = 0.64) but not in student-teacher relationships (*p* > 0.05). In the follow-up, no significant changes were observed in any of the variables, except for problem behaviors. However, the levels were maintained (*p* > 0.05). Only the problem behavior variable showed a significant decrease compared to the post-test (z = −2.03, *p* < 0.05, r = 0.64). In addition, in the post- and follow-up tests, children at risk of EBD reached performance levels similar to those of TD children in terms of all variables (*p* > 0.05), which is essential evidence showing the effectiveness and social validity of the intervention.

## 1. Introduction

Self-regulation, a life skill, refers to the skills and processes related to directing, planning, and controlling attention, cognition, emotion, and behavior in line with a goal ([50]; [78]). Although self-regulation skills begin to develop at birth, they are based on external regulation in infancy. In this process, when infants provide clues that they need regulation, their parents or primary caregivers regulate their needs. As children grow and mature, there is a shift from external to internal regulation. Factors such as children’s maturation, the development of the prefrontal cortex and its activation of other regions of the brain, the secure relationship children have with their caregivers, the presence of peers and adults who model self-regulation skills in their environment, and the opportunities children have to practice self-regulation skills play an important role in this change process ([7]; [25]; [37]).

The development of the dynamic structure of the brain’s functional networks, which are closely related to self-regulation, undergoes significant changes in early childhood, and these changes directly affect the brain’s information processing capacity. In [64]’s ([64]) study, significant age-related increases were observed in the temporal flexibility values of functional networks, such as executive control, salience, and default mode networks, related to self-regulation. This indicates that the networks gain more flexible structures over time, and the capacity to reorganize functional connections increases, reflecting the development of self-regulatory functions, such as attentional control and cognitive flexibility. The increase in the spatiotemporal diversity of the salience network with age is associated with self-regulatory skills, such as emotional awareness, switching between emotional states, and contextualizing behaviors. Childhood abuse is an important factor that affects the functional connectivity between brain networks. [12] ([12]) found that abuse may affect functional connectivity patterns in networks related to attention and emotion regulation in the long term. However, positive parenting may be a protective factor. In conclusion, the findings of these studies suggest that dynamic changes in the brain’s functional networks may form the neurological basis of self-regulation.

Children at risk of emotional and behavioral disorders (EBDs) have difficulties in self-regulation skills, that is, in controlling and regulating their emotions, behaviors, and attention. Children with aggressive behaviors in the preschool period exhibit low performance in inhibitory control skills ([53]), and the frequency of anger and antisocial behaviors seen in this period is associated with poor executive control ([31]). Effortful control is closely related to the concept of attention and emotion regulation. It is a structure that allows individuals to stop dominant reactions in favor of non-dominant but goal-related actions and to exercise conscious control over their behaviors ([1]). Due to their low levels of effortful control, children at risk of EBD are likely to reflect their emotions in a negative way ([14]; [23]).

Children at risk of EBD have low participation in learning activities and the ability to establish positive relationships with those around them, which affects their learning process ([29]; [61]). In addition, these children have difficulty understanding social situations and communicate less with the people around them ([28]), have difficulty recognizing situations where they feel stress or anger and managing their emotional reactions to these situations ([77]), and may perceive the behaviors of those around them as hostile due to their difficulties in interpreting social cues ([16]; [20]). However, the literature emphasizes that children with improved self-regulation skills in the preschool period are more competent in terms of social skills ([4]; [19]), can establish better relationships with their peers and teachers ([42]; [75]), and are more academically successful ([9]; [35]; [36]). Therefore, supporting self-regulation skills in the preschool period is very important in terms of preventing many possible problems that children at risk of EBD may experience.

Many comprehensive intervention programs in the literature aim to support preschool children’s social-emotional development and reduce their problem behaviors. Although these programs are designed as social-emotional or school readiness intervention programs in terms of their general structure, they include various goals to support the development of self-regulation skills. Such programs can intervene in the problem behaviors of children at risk while taking universal measures for children with typical development ([52]). In addition, these comprehensive intervention programs require teacher or parent training and include various, intervention-specific materials. Incredible Years, the most well-known of these interventions, is a comprehensive intervention consisting of parent, teacher, and child training programs. The program aims to reduce the impulsive behaviors of children with externalized behaviors and improve their self-control, self-esteem skills, and social, emotional, and academic competencies. In this context, the content of this intervention program includes self-regulation skills such as controlling anger, interpersonal problem-solving, and recognizing and understanding emotions. Many studies have shown that the program effectively reduces children’s aggressive and destructive behaviors with externalized behavior problems in early childhood ([70]). The Promoting Alternative Thinking Strategies (PATHS) program aims to develop children’s awareness of their own and others’ emotions, support the acquisition of a positive self-image, and teach self-control and problem-solving skills. One skill highly associated with self-regulation is the ability to solve interpersonal problems in constructive ways. The most well-known program for developing preschool children’s interpersonal problem-solving skills is the I Can Solve Problems program. The program primarily aims to reduce and prevent high-risk behaviors in preschool by teaching children how to think when solving interpersonal problems ([51]). The literature states that children participating in the program can produce more solutions to problems in the problem-solving process, increase adaptive behaviors, and decrease impulsive and externalized behaviors ([32]).

The literature shows that comprehensive school readiness intervention programs include early literacy skills, prosocial behaviors, and self-regulation skills. The Chicago School Readiness Project, which requires teacher training, examined the self-regulation and school readiness skills of 602 children in 35 Head Start classes over one year. The results of the study revealed that attention, inhibitory control, and executive functioning skills were developed for self-regulation, and word, letter naming, and math skills were developed for school readiness ([54]). The Kids in Transition to School (KITS) intervention program, which includes both child and family training, includes early literacy skills, prosocial behaviors, and self-regulation skills, such as coping with frustration, impulse and attention control, and following instructions. In their study, [47] ([47]) implemented the KITS program for children with externalizing behaviors. As a result of the research, it was found that there was an improvement in the positive behaviors and self-regulation skills of the children in the experimental group who participated in the program compared to the children in the control group.

When the results of the programs were examined, there was an increase in children’s positive behaviors and a decrease in their problem behaviors. However, various features, such as the content of these programs being quite extensive, requiring teacher and family training to gain implementation competence, having standard materials in the implementation process, and their financial burden, reduce the replicability of the programs in the classroom. Therefore, classroom-based interventions are needed in the literature. In addition, none of these intervention programs focused solely on developing self-regulation skills. This makes it difficult to determine which intervention component is related to the observed outcomes ([60]; [68]). Therefore, there is a need to examine the effects of interventions that directly target only improving self-regulation skills.

The literature states that there are a limited number of self-regulation intervention studies in the preschool period ([6]; [43]; [46]). Similarly, there are minimal intervention studies on self-regulation skills in the preschool period in Türkiye, and these studies are conducted only with typically developing (TD) children ([2]; [24]; [33]; [62]). Although self-regulation skills are critical for all individuals throughout life, they are also important for students at risk of EBD. It is expected that by supporting the self-regulation skills of children at risk of EBD, their positive social behaviors and participation in academic activities will increase, and accordingly, their problem behaviors will decrease, their social acceptance by their peers will increase, and their conflicts or problems with their teachers will decrease ([22]; [29]; [44]; [46]; [59]). If the problem behaviors exhibited by children at risk of EBD are not addressed in the preschool period, the behaviors exhibited will continue to increase ([39]; [55]). Therefore, these children need interventions that support their self-regulation skills.

For this reason, this study aimed to develop an intervention package that teachers can practically implement in their classrooms to support the self-regulation skills of children at risk for EBD and to examine the effect of this intervention on children’s behaviors. This study aimed to examine the effect of a self-regulation intervention package on self-regulation skills, problem behaviors, social skills, peer acceptance, and relationships with teachers of children at risk of EBD. In this direction, answers to the following research questions were sought: (a) Is there a significant difference between the pre-test and post-test scores of children with EBD on self-regulation, problem behaviors, social skills, peer acceptance, and teacher relationships? (b) Is there a significant difference between the post-test and follow-up scores of children with EBD risk for self-regulation, problem behavior, social skills, peer acceptance, and relationships with teachers? (c) Is there a significant difference between the pre-test scores of children with EBD risk and TD children in self-regulation, problem behavior, social skills, peer acceptance, and relationships with their teachers? (d) Is there a significant difference between the post-test scores of children at risk of EBD and TD in terms of self-regulation, problem behaviors, social skills, peer acceptance, and relationships with teachers? (e) Is there a significant difference between the follow-up scores of children at risk for EBD and TD in self-regulation, problem behavior, social skills, peer acceptance, and relationships with teachers?

## 2. Materials and Methods

### 2.1. Participants

A total of 10 children were included in the study, comprising five children at risk of EBD and five children with TD from their class for each child. The initial selection of children with both EBD and TD was based on the nominations of counselors and classroom teachers. Teachers were asked to nominate children in both groups who did not have a disability diagnosis. However, if the child at risk of EBD exhibited internalized or externalized behavioral problems for at least three months, this situation negatively affected the child’s educational performance and social relations. Children with TD were not referred in this respect, showing that these children were not at risk for EBD. While selecting TD children, teachers were asked to pay attention to children who did not have problems with peer acceptance and had good social skills. Some prerequisites were considered when determining whether the participants were at risk of EBD. The prerequisites were that the participants were between the ages of 60 and 72 months, did not have any diagnosis of disability, scored in the 90th percentile for their gender and age group in one of the anger-aggression or anxiety-withdrawal dimensions of the Social Competence and Behavior Evaluation-30 (SCBE-30) and in the social competence ([17]), had not participated in any intervention program for behavioral problems, and that the teacher and parent stated that the child had behavioral problems and that these problems negatively affected the quality of the child’s work and interaction with his/her environment. Further, the behavioral problems exhibited were observed by the teacher or parent for at least three months, and parents’ consent for their children’s participation in the study was obtained from the parents.

To identify children at risk of EBD, guidance counselors or classroom teachers were asked to nominate children with suspected risk of EBD. For the nominated children, the classroom teachers filled out the SCBE-30 ([17]). The 6-point Likert-type scale has three subscales: social competence, anger-aggression, and anxiety-withdrawal. Social competence assesses children’s positive characteristics, such as cooperation and finding solutions to problem situations; anger-aggression assesses externalized behavioral problems, such as defiance against adults and maladaptive and aggressive behavior in peer relationships; and anxiety-withdrawal assesses children’s internalized behavioral problems, such as timid behavior in groups and depressive moods. Children who scored in the 90th percentile in their sex and age group in one of the anger-aggression or anxiety-withdrawal dimensions of this scale and the social competence subscale were considered at risk for EBD. After the SCBE-30 was administered, functional assessment interviews were conducted with teachers and parents of children at risk of EBD. These interviews aimed to collect more detailed information about children’s behavioral issues. The interviews included various questions about the characteristics of the behavior, time of onset, duration of persistence, and its effects on educational performance and social relationships. As a result of the functional assessment, interviews determined that the child consistently exhibited behavioral problems at school and home, that the problem behaviors had been observed for at least three months, and that these behaviors negatively affected the child’s educational performance and relations with peers. Classroom observations were made for children who met these conditions for one school day, and anecdotal records were maintained. Five participants who met the prerequisites and were at risk of EBD were identified at the end of all processes. The demographic and behavioral characteristics of the participants are summarized in Appendix A.

### 2.2. Research Model

This study used a one-group pretest-posttest design. In Türkiye, children with EBD cannot officially benefit from special education services. In addition, children at risk of EBD are not systematically screened or monitored in schools. Therefore, the difficulty in identifying children at risk of EBD is an important factor in selecting this model. Since it was challenging to match children at risk for EBD in terms of behavioral problems, a single-group pretest-posttest model was preferred over a control group pre-test−post-test model. In addition, since the data collection tools used in the study did not allow for continuous data collection, this model was chosen rather than a single-subject experimental design. The literature states that this model is widely used to evaluate the effects of behavioral interventions ([15]). In this model, measurements of the dependent variables are taken from all participants before and after the intervention. The measurement results obtained are compared, and the significance of the difference between the results is examined. If there is a statistically significant difference between the results, it can be said that this difference is due to the intervention ([11]).

Various measures were taken to control for internal validity. The first researcher conducted the study with 10 participants continuing their education in the same school, so all participants were exposed to the same intervention conditions in the same environment. The intervention process lasted for two months. Thus, the maturation effect was controlled for. In addition, the teachers were not informed about the details of the intervention before or during the intervention, and were asked to continue their usual education processes. Thus, other variables that may have impacted the intervention results were controlled. In addition, all test conditions were carried out in the same environment, with the same materials, and by the same person, as planned, ensuring high implementation reliability. Thus, problems arising from the test conditions, instruments, and scoring were prevented.

### 2.3. Dependent and Independent Variable

The dependent variables of this study were self-regulation, problem behavior, social skills, peer acceptance, and relationships with teachers of children at risk for EBD. The independent variable of this study was the intervention package prepared to improve the self-regulation skills of children at risk for EBD. In the content of the intervention package, self-regulation skills such as paying attention to the object, situation, or event, problem-solving, acting according to the instructions, showing emotions in appropriate ways, explaining the emotions of others, self-motivation, taking responsibility, and making a plan for a purpose were targeted, and play-based activities were organized to achieve these goals.

### 2.4. Measures

#### 2.4.1. Preschool Self-Regulation Scale

The scale developed by [63] ([63]) was adapted into Turkish by [67] ([67]). The scale, which consists of two main parts, includes a practitioner guide regarding the tasks children are expected to perform and a practitioner evaluation form based on practitioner-child interaction. In the first part of the scale, the tasks of packing toys, waiting for toys, storing candy, and holding candy on the tongue were used to measure children’s delay of gratification levels. Balance board, tower building, and pencil clicking tasks determine children’s executive attention levels. Tower collection, toy sorting, and toy return tasks were used to measure children’s social adaptation skills. The second part of the scale, the practitioner evaluation form, assesses children’s attention, emotions, and behaviors based on practitioner-child interaction during the tasks. The practitioner evaluation form consists of 16 items: 10 in the attention/impulse control factor and 6 in the positive emotion factor. Each item was scored between 0 and 3. The reliability coefficients of the scale were 0.88 for the attention/impulse control factor, 0.80 for the positive emotion factor, and 0.83 for the whole scale.

#### 2.4.2. Kindergarten and Preschool Behavior Scale

This scale, developed by [41] ([41]) and revised in 2003, was designed to measure the social skills and problem behaviors of children aged 3–6 years. The scale was adapted into Turkish by [45] ([45]). The social skills scale consists of three sub-factors: social cooperation, social independence, and acceptance and social interaction, with a total of 23 items. The Cronbach’s alpha internal consistency coefficients for the sub-dimensions of the scale were 0.92, 0.88, and 0.88, respectively, while the total Cronbach’s alpha value was found to be 0.94. The problem behavior scale has four sub-factors: externalizing problems, internalizing problems, antisocial and egocentric problems, and a total of 27 items. The Cronbach’s alpha internal consistency coefficients for the sub-dimensions of the scale were 0.95, 0.87, 0.81, and 0.72, respectively, while the total Cronbach’s alpha value of the scale was found to be 0.96.

#### 2.4.3. Sociometry

Sociometry based on the peer rating method was applied in the study’s pre-test, post-test, and follow-up phases to evaluate the participants’ peer acceptance. The peer rating technique has several advantages over the peer nomination technique. In the peer nomination technique, children are actively encouraged to nominate peers they dislike, whereas in peer rating, children are asked to rate how much they enjoy spending time with each peer in their class. Thus, it reveals how children feel about all their peers in the classroom ([74]).

#### 2.4.4. Student-Teacher Relationship Scale-Short Form

The short form of the scale developed by [48] ([48]) was adapted into Turkish by [3] ([3]). The scale has two sub-dimensions: conflict and closeness. The conflict sub-dimension consists of 8 items, including the negative interaction between the teacher and the child, the teacher’s perception of the child’s behavior as unfavorable, and the teacher’s inability to manage these behaviors. The closeness sub-dimension consists of 7 items, including the teacher’s responsiveness toward the student and positive interaction with the student. In this study, data were collected by filling out the scales of the participants’ teachers. The internal consistency reliability coefficients were 0.82 for the whole scale, 0.84 for the conflict sub-dimension, and 0.76 for the closeness sub-dimension.

#### 2.4.5. Social Validity Parent-Teacher Interview Form

To obtain social validity data on the effectiveness of the intervention package, the researchers prepared the “Social Validity Parent and Teacher Interview Form.” Through semi-structured interviews, the views of parents and teachers were obtained regarding the effects of the activities in the intervention package on children’s self-regulation skills, problem behaviors, social skills, peer acceptance, and relationships with their teachers.

### 2.5. Self-Regulation Intervention Package

#### 2.5.1. The Theoretical Structure of the Intervention Package

The activities in the intervention package were prepared by adopting a play-based approach. Guided play is a teaching approach “between direct instruction and free play” ßwith a certain level of adult participation and learning goals in areas where the adult needs guidance. It includes fun and child-directed elements ([72]).

#### 2.5.2. Contents of the Intervention Package

The process of developing the intervention package is described in the following stages.

Identifying core principles. To guide the development and implementation of the activities, some basic principles were adopted in accordance with the theoretical structure of the intervention package. In line with these basic principles, it was ensured that the selected goals were the skills that children at risk of EBD have difficulty with in the field of self-regulation, that the goals were distributed in a balanced number, and that the activities were prepared in a spiral manner to allow for repetition of the goals. The activities were planned to support the interaction of children with EBD and TD with each other and to contribute to the formation of collaborative working environments. Voluntary participation of children in the activities was prioritized, and children who did not want to participate were allowed to watch the game for a certain period. Activity durations were organized according to the children’s interests. The first researcher implemented these activities. In this process, the researcher assumed the role of the person who prepared the learning environment for the children and guided the learning process with questions or various comments.Determining the objectives. While creating the content of the intervention package, the literature was reviewed, and the problems that preschool children at risk of EBD experience in self-regulation were identified. In addition, the characteristics related to self-regulation skills expected to be exhibited during the preschool period were determined based on the literature. The Preschool Education Curriculum in Türkiye was examined in terms of characteristics related to self-regulation skills during the preschool period. The existing goals related to self-regulation in the Turkish Preschool Curriculum were preserved, but the characteristics that existed in the literature but were not included in the curriculum were added as additional goals by the researchers. Expert opinions were obtained to evaluate the appropriateness of the goals determined for the research and the children’s developmental levels. The goals of the intervention package were created by deciding in line with the opinions. These goals are paying attention to the object/situation/event, producing solutions to problem situations, acting by the instruction or rule, explaining the feelings of others about a situation, showing positive/negative feelings about a situation in appropriate ways, motivating oneself to accomplish a task, taking responsibility, and planning for a purpose.Creating the activities. The researchers prepared the activities. The contents of intervention programs to improve preschool children’s self-regulation skills were examined ([2]; [5]; [24]; [37]; [60]; [62]; [68]; [71]). While preparing the activities, sample activities from the self-regulation literature and the Turkish Preschool Curriculum were examined. A total of 16 play-based activities were written in line with the developmental characteristics of children, the basic principles adopted, and the goals set, in a way that would allow each goal to be repeated at least five times in different sessions. The intervention package was created by obtaining expert opinions on the compatibility of the activities with the goals and their suitability for the developmental characteristics of the children. A sample activity plan is presented in Appendix A.

### 2.6. Procedure

#### 2.6.1. Pilot Study

The pilot study was conducted in a kindergarten different from the intervention group, with two different children at risk of EBD who met the prerequisites of the study. These two participants continued their education in the same class. Therefore, the pilot study was implemented as a large group with these two participants and TD children in their class. The pilot study was administered as a pre-test, intervention, and post-test, similar to the main study design. In the pilot study, all 16 play-based activities were implemented, and the appropriateness of the activities was evaluated. After the pilot study, material and instructional arrangements were made for some activities, and the intervention package was finalized.

#### 2.6.2. Pre-Test

After identifying five children at risk for EBD and one TD child for each at-risk child in a different preschool from the pilot study school, pre-test data were collected for all participants on the variables of self-regulation, problem behavior, social skills, peer acceptance, and student-teacher relationship. Data on problem behavior, social skills, and student-teacher relationship variables were collected using scales given to each participant’s teacher. The first researcher explained the purpose of the scales to the teachersand how to fill them out. In addition, the researcher asked the teachers to fill out the scales in a quiet environment and think about the students’ behaviors over the last three months.

The self-regulation variable was measured using the Preschool Self-Regulation Scale. The first researcher implemented a performance-based scale. To gain competence in implementing the scale, the first researcher received face-to-face online training on implementing the scale from one of the researchers who had adopted it. Before training, the researcher read the implementation instructions prepared by [67] ([67]) and watched the implementation videos. During the training, the researcher who adapted the scale explained the general structure of the scale and the implementation of the tasks and answered the researcher’s questions. After the training, the researcher provided all the materials to be used during the tasks, made a sample implementation with a child with TD who was not included in the study, and recorded the implementation on video. The video and the practitioner evaluation form, in which the child’s performance was scored, were sent to the researcher who provided the training. The training researcher examined the video and the form and provided feedback to the first researcher, who could then implement this scale. The implementation was carried out in a quiet environment in the educational institutions where the children attended, and one-on-one sessions with each child. After all the tasks were completed and the child left the assessment environment, the first researcher evaluated the child’s performance on emotion, behavior, and attention regulation skills using the practitioner evaluation form based on the scores she recorded regarding the child’s performance during the tasks.

Peer acceptance was measured using sociometry based on peer ratings. The first researcher implemented the sociometry process. The implementation process was carried out individually for each child in a quiet environment in the educational institution. The children were told that they would play a game about photographs, preference boxes were introduced, and the game’s rules were explained. The children were told, “Now I will show you the photos of your friends. First, you will tell me who your friend is in the photo. Then, if you prefer to play games with this friend, we will put the photo in the smiley face box; if you do not prefer to play games, we will put the photo in the sad face box, and if you are undecided/not sure about playing games, we will put the photo in the neutral face box. “To better understand the game, the researcher modeled the process through an example and then proceeded to the primary implementation. In the rating of peer preferences, the child was given 1 point if they put the photo in the sad face box, 2 points if they put it in the neutral face box, and 3 points if they put it in the smiley face box, and these were recorded on the sociometric rating scale. After the implementation with all the children in the class, to calculate each child’s sociometric rating score, the scores received from their peers were summed and divided by the number of peers who rated, and the average was taken.

The first researcher conducted the Preschool Self-Regulation Scale and sociometry implementation one-on-one with the children in an individual classroom within the school. The children did not experience any stranger effects from the researcher during this process. In identifying children at risk of EBD, the researcher made a full-day classroom observation in the classroom of the children and met the students by participating in the activities offered by the teacher.

#### 2.6.3. Intervention

The first researcher conducted the intervention in an empty classroom in a kindergarten in Ankara. Five children at risk of EBD attended various classes in this kindergarten, and one TD child from each child at risk of EBD was selected. Five TD children participated in this intervention. This intervention package, which was carried out with 10 children, consisted of 16 different play activities. Each activity was organized into a separate session. The 16-session intervention package was organized as one-hour sessions, two days a week for eight weeks. Before each session, the first researcher prepared the environment and materials to align with the goal and activity, and brought the children into the environment. In each session, she introduced herself to attract the children’s attention. She explained the game by establishing interactive dialogues with the children through the material or game she brought with her. Children’s voluntary participation in the game was prioritized, and those who did not want to participate were neither forced nor offered an external reinforcer. The games were fun and engaging, ensuring children’s participation. At the end of the session, the researcher emphasized the goals achieved in the game and concluded the session by thanking the children. The sessions could not be recorded with a camera for ethical reasons. However, photographs of the products or the process were taken so that the faces of the participants could not be seen.

#### 2.6.4. Post-Test and Follow-Up

After the intervention, post-test data on the dependent variables were collected from all participants. Three weeks after the end of the intervention, data on the dependent variables were collected in the same manner as in the pre-test and post-test to examine whether the effects of the intervention continued.

### 2.7. Reliability

In this study, implementation reliability data regarding the implementation processes of sociometry, self-regulation scale, and intervention package were collected by two experts who continued their doctoral education in special education. The doctoral student who conducted the implementation reliability of the intervention package was an experienced researcher who continued her doctorate in special education and worked as a scholar on various projects. Before the implementation of reliability, the researcher was informed about the general principles and structure of the intervention package. In addition, the implementation reliability forms, including the steps in the sessions she would observe, were provided in advance, and any questions were answered. The observer attended five intervention sessions. The implementation reliability of the intervention package was found to be between 91% and 100%.

Another doctoral student was utilized to collect the implementation reliability and inter-observer reliability data related to the implementation process of the sociometry and self-regulation scale. This student had participated in many projects and had previously carried out the sociometry process with children. Before implementation, the general structure of the self-regulation scale, the implementation reliability form, including the implementation steps, and the scoring process of the scale were explained to the doctoral student. Afterward, the consistency of the scoring was ensured through a sample video taken by the first researcher. Forms showing the sociometry implementation steps and a scoring chart were provided for sociometry. The implementation reliability of the Sociometry and Self-Regulation Scale was found to be 100%.

This doctoral student participated in 100% of the sociometry implementation’s pre-test, post-test, and follow-up data, and the inter-observer reliability was also 100%. The same observer participated in 100% of the pre-test data, 70% of the post-test data, and 40% of the follow-up data for the self-regulation scale implementation. At this point, the inter-observer reliability was 100%.

### 2.8. Data Analysis

Parametric tests require assumptions about specific population parameters, such as normal distribution, and test hypotheses regarding these parameters. In cases where these assumptions are not met, various hypothesis testing techniques called nonparametric tests are used as alternatives to parametric tests. This study used nonparametric statistics to consider the number of participants ([26]). The Wilcoxon test was used to examine the differences in related samples, and the Mann−Whitney U test was used in unrelated samples. The r coefficient was calculated for the effect size ([58]). The data obtained from the interviews for social validity were analyzed using content analysis.

## 3. Results

The pre-test, post-test, and follow-up test scores of all participants on the dependent variables are shown in Appendix A. Based on these scores, an increase in self-regulation, social skills, and peer acceptance variables, as well as a decrease in the problem behavior variable from pre-test to post-test for children at risk of EBD, was observed. Similarly, children with TD showed an increase in self-regulation, social skills, and peer acceptance variables and a decrease in the problem behavior variable between the pre-test and post-test stages. However, no changes were observed in children who scored high in the pre-test stage for positive variables or low in the pre-test stage for problem behavior variables. Changes in the student-teacher relationship variables were generally low in both groups.

In general, the self-regulation scores of children at risk of EBD in the pre-test stage were lower than those of their peers with TD. At this point, the second participant scored high in his group, while the eighth participant scored low. It can be said that the second participant scored high on the self-regulation scale because she was a student with internalized behavioral problems, but did not have any problems in terms of attention and impulse control. In contrast, the second participant was an average child in terms of social skills and peer acceptance, but his self-regulation skills needed support.

Appendix A shows the pre-test, post-test, and follow-up test results for the variables of self-regulation, social skills, problem behaviors, peer acceptance, and relationship with teachers of children with EBD risk and TD children.

Based on these findings, it was observed that the self-regulation, social skills, peer acceptance, and closeness dimension averages of the student-teacher relationship scale for children at risk of EBD were lower than those for TD children regarding the relevant variables. The problem behavior and conflict dimension averages of the student-teacher relationship scale were high. When the post-test and follow-up data were examined, an increase was observed in the self-regulation, social skills, peer acceptance, and closeness dimensions of the student-teacher relationship averages of both EBD and TD children, and a decrease was observed in the averages of problem behavior and conflict dimensions of the student-teacher relationship.

The Wilcoxon test was applied to examine the difference between pre-test and post-test scores of children at risk of EBD regarding self-regulation, problem behavior, social skills, peer acceptance, and student-teacher relationships. The results are presented in Appendix A.

Based on these findings, the difference between the pre-test and post-test scores of children with EBD risk for self-regulation, problem behavior, social skills, and peer acceptance was statistically significant (z = −2.02, *p* < 0.05). The effect size for this difference (r = 0.64) was high. The median of the post-test scores (40.00) for the self-regulation variable was greater than the median of the pre-test scores (28.00), and the median of the post-test scores (37.00) for the problem behavior variable was smaller than the median of the pre-test scores (72.00). The median of the post-test scores for the social skills variable (78.00) was higher than the median of the pre-test scores (64.00), and the median of the post-test scores for the peer acceptance variable (2.00) was higher than the median of the pre-test scores (1.85). The difference between the pre-test and post-test scores obtained in terms of the conflict dimension of children’s relationships with their teachers (z = −1.60, *p* > 0.05) and the difference between the pre-test and post-test scores obtained in terms of the closeness dimension (z = −1.84, *p* > 0.05) were not statistically significant.

The Wilcoxon test results regarding the difference between the post-test and follow-up test scores of children at risk of EBD for the relevant variables are shown in Appendix A.

Based on these findings, the difference between the post-test and follow-up scores obtained by children with EBD risk for self-regulation, social skills, peer acceptance, and relationships with teachers (conflict and closeness dimensions) was not statistically significant (*p* > 0.05). Only for the problem behavior variable, the difference between the post-test and follow-up scores was statistically significant (z = −2.03, *p* < 0.05). The effect size for this difference (r = 0.64) is at a high level. The median of the follow-up test scores (32.00) was smaller than the post-test scores (37.00). This shows that the problem behaviors of children at risk for EBD continued to decrease significantly from the post-test to the follow-up test.

It can be said that children at risk of EBD made progress in the pre-test, post-test, and follow-up tests in terms of relevant dependent variables. However, there is a need to compare the pre-test, post-test, and follow-up test performances of children at risk for EBD and TD children. The Mann–Whitney U test was used for this purpose. The results are presented in Appendix A.

Based on these findings, there was no statistically significant difference (*p* > 0.05) between children at risk for EBD and TD children in terms of self-regulation, problem behavior, and student-teacher relationships. However, the mean scores for children at risk for EBD were lower for these variables. On the other hand, it was observed that children at risk for EBD differed significantly from TD children in terms of social skills (U = 1.50, z = −2.31, *p* < 0.05) and peer acceptance (U = 0.00, z = −2.61, *p* < 0.05) variables. The effect size for the difference was high for social skills (r = 0.73) and peer acceptance (r = 0.83). The median score of the social skills variable of the children with EBD risk was (64.00), it was (85.00) for the TD children. In the peer acceptance variable, the median score of the children at risk of EBD was (1.85), while that of the TD children was (2.35). The post-tests of children at risk of EBD and TD children regarding the relevant variables are compared in Appendix A.

Based on these findings, there was no statistically significant difference between children at risk of EBD and TD children in terms of self-regulation, problem behavior, social skills, peer acceptance, and student-teacher relationships in the post-test phase (*p* > 0.05). This shows that with the effect of the intervention package, children at risk of EBD reached a similar performance level to TD children in the related dependent variables. A follow-up test was conducted to evaluate whether this effect achieved in the post-test continued after three weeks. A comparison of the follow-up tests for children at risk of EBD and TD children is shown in Appendix A.

Based on these findings, no statistically significant difference existed between the follow-up test scores of children at risk for EBD and TD children (*p* > 0.05). This shows that children at risk of EBD reached and maintained performance levels similar to TD children in the related variables.

### Social Validity

After the post-tests, social validity data were collected via individual interviews with the participants’ parents and teachers. Parent interviews were conducted with parents of participants at risk for EBD and parents of TD participants, while teacher interviews were conducted only for participants at risk of EBD. Since teachers nominated children with TD for their good social skills, no teacher social validity interviews were conducted with these children. All the interviews were audio recorded. According to the content analysis results, all parents reported significant improvements in their children’s self-regulation skills. In addition, all parents of children at risk for EBD stated that there was a significant decrease in their children’s problem behaviors. While nine out of ten parents reported improvements in their children’s social skills and increased peer acceptance, one parent of one of the EBD participants reported that their child already had good social skills. Only two of the ten parents reported positive changes in the student-teacher relationship. As a result of the interviews with the teachers, all the teachers stated that children at risk of EBD acquired various self-regulation skills, their problem behaviors decreased, they gained various social skills, and there were positive developments in peer acceptance and friend relations, albeit at different levels. However, different responses were received regarding the reflections of the intervention package on the student-teacher relationships. While some teachers stated that there was an increase in closeness in their relationship with their students at risk of EBD, some teachers stated that there was no change.

## 4. Discussion

Within the study’s limitations, the intervention may be effective in the self-regulation, social skills, peer acceptance, and problem behaviors of children at risk for EBD. However, it was not effective in their relationships with teachers. The content of the intervention package included objectives such as focusing, maintaining and controlling attention, inhibitory control, controlling negative emotions, or acting by the instructions and rules, which are various, comprehensive, and compatible with the intervention contents in the literature, which may explain the overlap of the effectiveness findings of this intervention with the literature ([5]; [21]; [47]; [54]). It is thought that the most crucial reason for the lack of a significant change in the student-teacher relationship variable is that the implementation was carried out in an environment different from the child’s classroom environment and by a practitioner different from the teacher.

In the pre-test phase, there was no significant difference between children at risk of EBD and TD children in terms of self-regulation, problem behaviors, or their relationships with their teachers (*p* > 0.05). Although this situation suggests that both groups have similar performance levels in these aspects, descriptive statistics results show that the mean scores of children at risk of EBD in terms of self-regulation and closeness with teachers were lower than those of TD children. In comparison, the mean scores for problem behavior and conflict with teachers were higher. In the literature, the self-regulation performance of children with EBD or at risk for EBD is low ([31]; [53]; [65]). Students with ADHD, which is associated with EBD, are behind their TD peers in areas such as inhibitory control, working memory, and short-term memory related to behavioral self-regulation or executive functions ([65]), and children with aggressive behaviors in the preschool period exhibit lower performance in inhibitory control skills ([53]). The frequency of anger and antisocial behavior exhibited by children with disruptive behaviors during this period is related to poor executive control ([31]). In this study, there were no differences between the two groups in terms of self-regulation, problem behaviors, or student-teacher relationships. This can be explained by the fact that there were five participants in each group and the individual characteristics of the children. In addition, during the selection process of the TD children, peers who had fewer problem behaviors but were competent in terms of social skills were to be determined based on the teacher’s statement. Therefore, considering subject subjectivity, children who did not have a predominant problem behavior among TD children, such as Participant Eight, who performed averagely in social skills but had low self-regulation skills, may have been selected. Similarly, among the students at risk of EBD, the presence of students who, like the second participant, had dominant introversion but average or above-average performance in self-regulation skills may have led to this result.

The activities carried out for the objectives were play-based, which may have increased the children’s self-regulation performance. Play is a valuable tool that provides the appropriate context for supporting self-regulation skills and allows children to control their mental activities ([10]; [69]). The implemented intervention package consisted of fun and collaborative activities that may have supported children’s ability to express their ideas, explain their reasoning processes, and talk about their learning processes. These important skills may have improved children’s self-regulation ([49]).

Features such as the practitioner’s approach, how she presented the activities, and the language of communication she used with the children may have also increased the children’s self-regulation performance. In the process, the practitioner provided children with opportunities to explain their thoughts and thinking processes, model thinking processes or strategies by thinking aloud, give children opportunities to practice, and provide feedback to them. In the literature, it is stated that all practices, such as adults giving children the opportunity to make choices and express their ideas clearly ([13]; [27]), modeling related processes by thinking aloud ([25]; [73]), giving children opportunities to practice, providing support at the level needed, reinforcing successful attempts, and gradually withdrawing the support provided, all support self-regulation skills ([25]; [57]).

Although there was no significant difference between the two groups in terms of the problem behavior variable in the pre-test due to the small number of participants, the intervention applied in the post-test was effective, and the only variable that showed a significant difference between the post-test and the follow-up test was problem behavior. Improvements in self-regulation skills may have been effective in reducing children’s behavioral problems. Many studies have shown that self-regulation skills can effectively prevent and reduce behavioral problems ([8]; [38]; [56]). There is an opposite relationship between problem behavior and social skills variables ([32]; [70]). It is expected that the social skills of a child whose problem behaviors decrease will also increase. In the pre-test phase, there was a significant difference between the two groups regarding the social skills variable (*p* > 0.05), and in the post-test and follow-up tests, children at risk of EBD reached a similar performance level as their TD peers. As a result, children whose self-regulation skills improved with the intervention package were expected to increase their social skills. In the literature, it is seen that children with improved self-regulation skills have advantages in terms of social relationships ([40]), social skills ([4]), and social competence ([19]).

In the pre-test phase, the acceptance of children at risk of EBD by their peers was significantly lower than that of their TD peers. This finding supports the literature that children at risk of EBD often have problems with their peers and are rejected by them ([42]; [66]). After the intervention, children at risk of EBD reached a similar level of peer acceptance as their TD peers. The development of self-regulation skills can explain this increase in peer acceptance. [30] ([30]) found that experiencing peer problems leads to lower levels of executive functioning, while higher levels of executive functioning reduce the likelihood of experiencing peer problems in childhood and middle adolescence.

In addition, sociometric status among peers is not an immutable result of individual characteristics or behavioral patterns; rather, it is a product of interactions between individuals and groups ([66]; [76]). Although preschool education environments are places where many different peers can interact with each other, the emergence point of these interactions is play. Studies have shown that play is especially effective in developing social competence and self-regulation skills; it supports children’s problem-solving skills, inhibits impulsive behaviors, expresses their emotions, and follows social rules ([18]; [34]). In this study, preparing the intervention package with a play-based approach and organizing the activities to support the interaction between children and contribute to forming collaborative working environments may have contributed to the significant increase in the peer acceptance scores of children at risk of EBD.

### Limitations and Future Research

The limited number of participants was the most critical limitation of this study. The results obtained due to the limited number of participants should be approached cautiously. It is essential to apply the intervention package to a larger group of participants in future studies to ensure the validity of the results.

The intervention package produced effective results for all dependent variables, except for the student-teacher relationship. Even though there was a decrease in the average scores of conflict in the student-teacher relationship and an increase in closeness scores, this was not significant. This result may be because the teacher must present the intervention package directly. For this reason, the effects of presenting the intervention package by the child’s teacher in order to increase the quality of the child’s relationship with the teacher can be evaluated in further studies. Similarly, in this study, the intervention package was conducted in an environment different from the children’s classroom environment and with children who were different from their classmates. The effects of conducting the intervention package with the child’s friends at risk of EBD in their classroom environment can be examined in further studies.

Another limitation of this study may be that TD children were selected based solely on the teachers’ opinions. In future studies, it is recommended that all dependent variables for these children be determined by setting various prerequisites and using various measurement tools.

## 5. Conclusions

In summary, many comprehensive social-emotional intervention programs have positive results for problem behaviors. In addition, the relationship between self-regulation and positive social behavior has been demonstrated. Although this intervention package, which directly targets the development of self-regulation skills, is not comprehensive, it has significantly contributed to the literature by reducing the problem behaviors of children at risk for EBD and improving their social skills and peer acceptance. In addition, it is important that the intervention package consists of play-based activities that can be easily implemented in the classroom and guide preschool teachers in its implementation.

## Data Availability

The raw data supporting the conclusions of this article will be made available by the authors upon request.

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
