# Peer review of "Self-Regulation Intervention Impact on Turkish Children with Emotional and Behavioral Disorder Risks"

_behavsci, 2025, doi:10.3390/bs15040508_

Round 1
Reviewer 1 Report
Comments and Suggestions for Authors
Thank you for your article. I enjoyed reading your research very much. I hope my comments help with further developing this work so it can be published soon.
Abstract - A tidy Abstract informing the reader of the main point of the study. Line 14: Superfluous words - can delete ‘As a result of the study’ (this is evident throughout the text so editing is needed). Line 12 – as above for ‘the fact’
Introduction – I enjoyed reading the Introduction as it provided a solid background to the topic and introduced the issues which need to be investigated. I thought that more up to date literature was needed to support the topic presented. There is some mor recent literature, but their appears to be an over-reliance of older literature. Of course, keep the seminal literature (eg Gresham and Blair) but update to more recent work.
Lines 31 - 33: There is a disconnect with the ideas in this sentence, thus I am unsure what the authors are meaning. What are effortful control skills? This is not a term I am familiar with. Does’ effortful ‘mean that a child can do a skills / mastered a skill or the opposite? I think this word needs to be replaced with a more overt word which the international audience will be familiar with.
Line 103 – delete superfluous words (and highlighted in yellow throughout text)
Materials and Methods
Participants: Lines 108 – 109 and line 117: Details needed via a small explanation /description of the selection tools. This will provide an overview of the style of questions asked etc.
Line 120 - this is the first mention of the Functional Assessment interview – why did they do this, what did they do, who with, and when, Details are missing and they need to be here.
Research Model. Lines 128-133 are vague. Could this be rewritten to provide a positive use of pre and post test. Comment - given the small number of children involved I would have thought a single case design would have shown more effective of the intervention.
Dependent and Independent Variables: Lines 135-141. Some of the variables were explained but what does self motivation, taking responsibility and making a plan with a purpose look like and how were these behaviours recorded?
It has not been explained how the first researcher built a bond with the chdilren (in order to do the measures). Details are missing on the actions taken.
Measures. Refer to text for comments and recommendations
The Intervention Package –Line 196. From the writing, I am unclear how the actual intervention package was decided. Can the authors make this clear and provide detail on how they selected the 16 play-based activities step by step.
Lines 202-204. Does this mean just the EBD children or the EBD and TD children played together – please make this clear. It was not until the end of the article that it was explained that the children did not know one another.
How were the play-based activities implemented, when, who with etc and for how long. Keep order in this and clearly explain the general process for each session. Currently this is unclear.
Pilot Study. Line 222. For a pilot, the target children should not be used. Do the authors mean different children or two of the five target children. The pilot study needs to be reported with more detail. What changes were made from the pilot study to the main study? Detail needed.
Procedures (lines 221)- through to Social Validity (line 265)
See comments on text
Social validity. See comments on text. I am lost as to whether the interviews were part of the assessment measures or just part of the social validity reporting. This section needs to be reviewed and revised accordingly to what /who did what. The findings of the social validity scale should be placed in the Findings section, not in Methods.
Results line 289 -355. See comments on text. Given the very small number of children participating, the pre-post test design appears inappropriate (ie combining five EBD together and 5 TD together). Given the measures, a different research design may have provided baseline and the effects of intervention more clearly for individual children and as a group. Overall though, I personally would have rather seen the children’s actual scores in Table format so I could compare across phases and also across children (both EBD and TD).
As presented, the pre-post tables are appropriate and clear to the reader.
Discussion lines 356 –423 see notes on text.
Line 386 – there is a statement about the second participant – no other participants have been mentions so this feel out of place. As noted on text and in this summary, the children’s data/findings need to be more personalised so the mention of a specific participant is understood by the reader.
Lines 402 -423. This paragraph has too many ideas running through it so it is confusing to read. Separate to make each point clear and relate to the findings and literature.
Limitations (lines 425-440) were appropriately identified as is further studies (not different).
A conclusion is missing and there needs to be one included.

See notes on text. There is use superfluous words and deleting these will make for easier reading. Some structuring of paragraphs will help also with readability of text (see comments on text and above). .
Author Response
Thank you very much for your feedback, and we believe that all of the feedback has improved the research considerably.
Comment 1: (Superfluous words - can delete ‘As a result of the study’ (this is evident throughout the text so editing is needed). Line 12 – as above for ‘the fact’)
Response 1: (Thank you for pointing this out. We agree with this comment. I have revised the abstract and the manuscript in general and eliminated superfluous words.)
Comment 2: Introduction – I enjoyed reading the Introduction as it provided a solid background to the topic and introduced the issues which need to be investigated. I thought that more up to date literature was needed to support the topic presented. There is some mor recent literature, but their appears to be an over-reliance of older literature. Of course, keep the seminal literature (eg Gresham and Blair) but update to more recent work.
Response 2: (Thank you for pointing this out. We agree with this comment. For this reason, we have left the pioneering sources as they are and added new studies to emphasize the subject and its importance with more up-to-date sources.
(The literature shows that the content of comprehensive school readiness intervention programs includes early literacy skills, prosocial behaviors, and self-regulation skills. The Chicago School Readiness Project, which requires teacher training, examined the self-regulation and school readiness skills of 602 children in 35 Head Start classes over a year. As a result of the study, it was revealed that attention, inhibitory control, and ex-ecutive function skills were developed about self-regulation, and word, letter naming, and math skills were developed about school readiness (Raver et al., 2011). The Kids in Transition to School (KITS) intervention program, which includes both child and family training, includes early literacy skills, prosocial behaviors, and self-regulation skills such as coping with frustration, impulse and attention control and following instructions. In their study, Pears et al. (2012) implemented the KITS program for children with exter-nalizing behaviors. As a result of the research, it was found that there was an im-provement in the positive behaviors and self-regulation skills of the children in the experimental group who participated in the program compared to the children in the control group. (p2 line 80-83)
It is stated in the literature that there are a limited number of self-regulation intervention studies in the preschool period (Baron et al., 2017; Morawska et al., 2019; Pandey et al., 2018). (p3 line 105-106).
Although self-regulation skills are critical for all individuals throughout life, they are also important for students at risk of EBD. It is expected that by supporting the self-regulation skills of children at risk of EBD, their positive social behaviors and participation in academic activities will increase and, accordingly, their problem behaviors will decrease, their social acceptance by their peers will increase, and their conflicts or problems with their teachers will decrease (Edossa et al., 2018; Hasty et al., 2023; O’Donnell et al., 2024; Pandey et al., 2018; Salerni & Messetti, 2025). Considering that if the problem behaviors exhibited by children at risk of EBD are not intervened in the preschool period, the behaviors exhibited will continue to increase (Meagher et al., 2009; Reef et al., 2011), it is seen that these children need interventions that support their self-regulation skills. (p3 line 110-119).
Comment 3: (Lines 31 - 33: There is a disconnect with the ideas in this sentence, thus I am unsure what the authors are meaning. What are effortful control skills? This is not a term I am familiar with. Does’ effortful ‘mean that a child can do a skills / mastered a skill or the opposite? I think this word needs to be replaced with a more overt word which the international audience will be familiar with.
Response 3: (Thank you for pointing this out. We defined efforful control with the current source.
Effortful control is closely related to the concept of attention and emotion regulation. It is a structure that allows the individual to stop dominant reactions in favor of non-dominant but goal-related actions and to exercise conscious control over their behaviors (André et al., 2019). (p1, line 32-35).
Comment 4: (Participants: Lines 108 – 109 and line 117: Details needed via a small explanation /description of the selection tools. This will provide an overview of the style of questions asked etc.
Response 4: (Thank you for pointing this out. We have included details about the content of the SCBE-30 and the functional assessment interview.)
The 6-point Likert-type scale has three subscales: social competence, anger-aggression, and anxiety-withdrawal. Social competence assesses children's positive characteristics, such as cooperation and finding solutions to problem situations; anger-aggression assesses externalized behavioral problems, such as defiance against adults and maladaptive and aggressive behavior in peer relationships; and anxiety-withdrawal assesses children's internalized behavioral problems, such as timid behavior in groups and depressive moods. (p.4, line 162-168).
After the SCBE-30 was administered, functional assessment interviews were conducted with the teachers and parents of children at risk of EBD. These interviews aimed to collect more detailed information about children's behavioral problems. The interviews included various questions about the characteristics of the behavior, time of onset, duration of persistence, and its effects on educational performance and social relationships. (p. 4, line 171-176)
Comment 5: (Research Model. Lines 128-133 are vague. Could this be rewritten to provide a positive use of pre and post test. Comment - given the small number of children involved I would have thought a single case design would have shown more effective of the intervention.)
Response 5: Thank you for pointing this out. We explained that we did not prefer a single-subject research design because the instruments used did not allow continuous data collection. We justified the appropriateness of the pretest-posttest model and explained the measures we took for internal validity.
Since it was challenging to match children at risk for EBD in terms of behavioral problems, a single-group pretest-posttest model was preferred instead of a control-group pre-test-posttest model. In addition, since the data collection tools used in the study did not allow for continuous data collection, this model was decided upon rather than a single-subject experimental design. The literature states that this model is widely used to evaluate the effect of behavioral interventions (Cranmer, 2017). In this model, measurements regarding the dependent variables are taken from all participants before and after the intervention. The measurement results obtained are compared, and the significance of the difference between the results is examined. If there is a statistically significant difference between the results, it can be said that this difference is due to the intervention (Büyüköztürk et al., 2018).
Various measures were taken to control internal validity. The first researcher con-ducted the study with 10 participants who were continuing their education in the same school, so everyone was exposed to the same intervention conditions in the same environment. The intervention process lasted for two months. Thus, the maturation effect was controlled. In addition, the teachers were not informed about the details of the intervention before and during the intervention and were asked to continue their usual education processes. Thus, other variables that may impact the intervention results were controlled. In addition, all test conditions were carried out in the same environment, with the same materials, and by the same person, as planned, with high implementation reliability. Thus, problems arising from test conditions, instruments, and scoring were prevented. (p.4-5, line 185-205).
Comment 6: (Dependent and Independent Variables: Lines 135-141. Some of the variables were explained but what does self motivation, taking responsibility and making a plan with a purpose look like and how were these behaviours recorded?)
Response 6: (Thank you for your comment, but we have not made any changes for this comment. Each of these behaviors is considered as a part of self-regulation skills. These behaviors were reached by examining the literature on self-regulation skills in preschool period and taking expert opinions. Therefore, the Preschool Self-Regulation Scale was used to assess self-regulation skills.)
Comment 7: (Measures. Refer to text for comments and recommendations. This belongs in procedure. Information is needed on the actual scale)
Response 7: (Thank you for pointing this out. We have expanded the information about the structure of the measurement tools, especially the reliability or validity information.
The scale, which consists of two main parts, includes a practitioner guide regarding the tasks children are expected to perform and a practitioner evaluation form based on the practitioner-child interaction. In the first part of the scale, the tasks of packing toys, waiting for toys, storing candy, and holding candy on the tongue are used to measure children's delay of gratification levels. Balance board, tower building, and pencil clicking tasks determine children's executive attention levels. The tower collection, toy sorting, and toy return tasks measure children's social adaptation skills. The second part of the scale, the practitioner evaluation form, assesses children's attention, emotions, and behaviors based on the practitioner-child interaction during the tasks. The practitioner evaluation form consists of 16 items: 10 in the attention/impulse control factor and 6 in the positive emotion factor. Each item is scored between 0 and 3. The reliability coefficients of the scale were .88 for the attention/impulse control factor, .80 for the positive emotion factor, and .83 for the whole scale. (p5, line 219-231)
The Cronbach's alpha internal consistency coefficients for the sub-dimensions of the scale were .92, .88, and .88, respectively, while the total Cronbach's alpha value was found to be .94. (p 5, line 237-239)
The Cronbach's alpha internal consistency coefficients for the sub-dimensions of the scale were .95, .87, .81, and .72, respectively, while the total Cronbach's alpha value of the scale was found to be .96. (p6, line 240-243).
The internal consistency reliability coefficients were .82 for the whole scale, .84 for the conflict sub-dimension, and .76 for the closeness sub-dimension. (p6, 260-262)
We have explained the administration of the measurement tools in the pretest sub-heading of the procedure heading. This heading also mentions the precautions taken.
After identifying five children at risk for EBD and one TD child for each at-risk child in a different preschool from the pilot study school, pre-test data were collected for all participants on the variables self-regulation, problem behavior, social skills, peer acceptance, and student-teacher relationship. Data on problem behavior, social skills, and student-teacher relationship variables were collected through scales given to each participant's teacher. The first researcher explained to the teachers the purpose of the scales and how to fill them out. In addition, the researcher asked the teachers to fill out the scales in a quiet environment and by thinking about the student's behaviors for the last three months.
The self-regulation variable was measured with the preschool self-regulation scale. The first researcher implemented the performance-based scale. To gain competence in implementing the scale, the first researcher received face-to-face online training on im-plementing the scale from one of the researchers who adopted it. Before the training, the researcher read the implementation instructions prepared by Tanrıbuyurdu and Güler Yıldız (2014) and watched the implementation videos. During the training, the researcher who adapted the scale explained the general structure of the scale and the implementation of the tasks and answered the researcher's questions. After the training, the researcher provided all the materials to be used during the tasks, made a sample implementation with a child with TD who was not included in the study, and recorded it on video. This video and the practitioner evaluation form in which the child's performance was scored were sent to the researcher who provided the training. The training researcher examined the video and the form and gave feedback to the first researcher, who could then implement this scale. The implementation was carried out in a quiet environment in the educational institutions where the children attended and one-on-one with each child. After all the tasks were completed and the child left the assessment environment, the first researcher evaluated the child's performance on emotion, behavior, and attention regulation skills with the practitioner evaluation form based on the scores she recorded regarding the child's performance during the tasks.
The peer acceptance variable was measured with a sociometry implementation based on peer ratings. The first researcher carried out the sociometry implementation process. The implementation process was carried out individually with each child in a quiet environment in the educational institution. The children were told that they would play a game about photographs, the preference boxes were introduced, and the game's rules were explained. The children were told, "Now I will show you the photos of your friends. First, you will tell me who your friend is in the photo. Then, if you prefer to play games with this friend, we will put the photo in the smiley face box; if you do not prefer to play games, we will put the photo in the sad face box, and if you are undecided/not sure about playing games, we will put the photo in the neutral face box. "In order to better understand the game, the researcher modeled the process through an example and then proceeded to the primary implementation. In the rating of peer preferences, the child was given 1 point if he/she put the photo in the sad face box, 2 points if he/she put it in the neutral face box, and 3 points if he/she put it in the smiley face box, and these were recorded on the sociometric rating scale. After the implementation with all the children in the class, to calculate each child's sociometric rating score, the scores received from his/her peers were summed and divided by the number of peers who rated, and the average was taken.
The first researcher conducted the preschool self-regulation scale and the sociometry implementation one-on-one with the children in an individual classroom within the school. The children did not experience any stranger effect on the researcher during this process. In identifying children at risk of EBD, the researcher made a full-day classroom observation in the classroom of the children and met the students by participating in the activities offered by the teacher. (line 331-382)
Comment 8: In Sociometry heading (Explain this measure clearly and reference it the highlight needs to be in Procedures)
Response 8: We have defined and referenced sociometrics based on peer rating. We have described the implementation of the sociometrics process in the pretest heading, the modification made in the previous comment is visible.
The peer rating technique has several advantages over the peer nomination technique. In the peer nomination technique, children are actively encouraged to nominate peers they dislike, whereas in peer rating, children are asked to rate how much they enjoy spending time with each peer in their class. Thus, it is revealed how children feel about all their peers in the classroom (Williams & Gilmour, 1994). (p6, line 246-251)
Comment 9: The Intervention Package –Line 196. From the writing, I am unclear how the actual intervention package was decided. Can the authors make this clear and provide detail on how they selected the 16 play-based activities step by step.
Response 9: The Contents of the Intervention Package describes the development of the intervention package in three steps. The process of developing the intervention package is described in stages.
- Identifying core principles. To guide the development and implementation of the ac-tivities, some basic principles were adopted in line with the theoretical structure of the intervention package. In line with these basic principles, it was ensured that the selected goals were the skills that children at risk of EBD have difficulty within the field of self-regulation, that the goals were distributed in a balanced number, and that the ac-tivities were prepared in a spiral manner to allow for repetition of the goals. The activities were planned to support the interaction of children with EBD and TD with each other and to contribute to the formation of collaborative working environments. Voluntary par-ticipation of children in the activities was prioritized, and children who did not want to participate were allowed to watch the game for a certain period. Activity durations were organized according to children's interests. The first researcher implemented the activi-ties. In this process, the researcher tried to assume the role of the person who prepared learning environments for children and guided the learning process with questions or various comments.
- Determining the objectives. While creating the content of the intervention package, the literature was reviewed, and the problems that preschool children at risk of EBD expe-rience in self-regulation were revealed. In addition, the characteristics related to self-regulation skills expected to be exhibited in the preschool period were determined in line with the literature. The Preschool Education Curriculum in Turkey was examined regarding characteristics related to self-regulation skills in the preschool period. The existing goals related to self-regulation in the Turkish Preschool Curriculum were pre-served, but the characteristics that existed in the literature but were not included in the curriculum were added as additional goals by the researchers. Expert opinions were obtained to evaluate the appropriateness of the goals determined for the research and the children's developmental level. The goals in the intervention package were created by deciding in line with the opinions. These goals are paying attention to the ob-ject/situation/event, producing solutions to problem situations, acting by the instruction or rule, explaining the feelings of others about a situation, showing positive/negative feelings about a situation in appropriate ways, motivating oneself to accomplish a task, taking responsibility and planning for a purpose.
- Creating the activities. The researchers prepared the activities. The contents of inter-vention programs to improve preschool children's self-regulation skills were examined (Arslan Çiftçi, 2020; Barnett et al., 2008; Ezmeci, 2019; McClelland & Tominey, 2015; Schmitt et al., 2015; Sezgin, 2016; Tominey & McClelland, 2011; Webster-Stratton & Reid, 2003). While preparing the activities, sample activities in the self-regulation literature and the Turkish preschool curriculum were examined. A total of 16 play-based activities were written in line with the developmental characteristics of children, the basic principles adopted, and the goals set, and in a way that would allow each goal to be repeated at least five times in different sessions. The intervention package was created by obtaining expert opinions on the compatibility of the activities with the goals and their suitability for the developmental characteristics of children. The sample activity plan is given in Table S2. (line 276-318)
Comment 10: Lines 202-204. Does this mean just the EBD children or the EBD and TD children played together – please make this clear. It was not until the end of the article that it was explained that the children did not know one another.
How were the play-based activities implemented, when, who with etc and for how long. Keep order in this and clearly explain the general process for each session. Currently this is unclear.
Response 10: All requested changes are described in the intervention heading.
The first researcher conducted the intervention in an empty classroom in an Ankara kindergarten. Five children at risk of EBD attended various classes in this kindergarten, and one TD child from each child at risk of EBD class was selected. Five TD children participated in the intervention. This intervention package, which was carried out with 10 children, consisted of 16 play activities. Each activity was organized as a separate session. The 16-session intervention package was organized as one-hour sessions two days a week for eight weeks. Before each session, the first researcher prepared the environment and materials to align with the goal and activity and brought the children into the envi-ronment. In each session, she introduced herself to attract the children's attention. She explained the game by establishing interactive dialogues with the children through the material or game she brought. Children's voluntary participation in the game was prioritized, and children who did not want to participate were not forced or offered an external reinforcer. The fact that the games were fun and engaging ensured children's participation. At the end of the session, the researcher emphasized the goals achieved in the game and ended the session by thanking the children. (p9, line 384-397).
Comment 11: Pilot Study. Line 222. For a pilot, the target children should not be used. Do the authors mean different children or two of the five target children. The pilot study needs to be reported with more detail. What changes were made from the pilot study to the main study? Detail needed.
Response 11: The pilot study is detailed as requested.
The pilot study was conducted in a kindergarten different from the intervention group, with two different children at risk of EBD who met the prerequisites. These two participants continue their education in the same class. Therefore, the pilot study was implemented as a large group with these two participants and TD children in their class. The pilot study was administered as a pre-test, intervention, and post-test, similar to the main study. In the pilot study, all 16 play-based activities were implemented, and the appropriateness of the activities was evaluated. After the pilot study, material and in-structional arrangements were made for some activities, and the intervention package was finalized. (p7, line 321-329)
Comment 12: In pretest heading (Where, when, who by etc - detail needed)
Response 12: All requested changes are described in the pre-test heading. (line 331-382)
Comment 13: In the post-test and follow up test, superfluous words were removed and the follow up test was asked to be done in the same way as the pre-test and post-test.
Response 13: All requested changes are described in the Post-Test and Follow-up heading. (Three weeks after the end of the intervention, data on dependent variables were collected in the same manner as in the pre-test and post-test to examine whether the effects of the intervention continued. (p9, line 403-405).
Comment 14: In the Reliability heading (who were doctoral students and Keep order and remove superfluous text)
Response 14: Doctoral students were explained who they are with their competencies. In addition, it was explained how they were trained by the researcher before collecting the reliability data.
(The doctoral student who conducted the implementation reliability of the activity package is an experienced researcher who continues her doctorate in special education and has worked as a scholar on various projects. Before the implementation reliability, this re-searcher was informed about the general principles and structure of the activity package. In addition, the implementation reliability forms, including the steps in the sessions she would observe, were given in advance, and any questions were answered. The observer attended five intervention sessions. The implementation reliability of the activity package was found to be between 91% and 100%.
Another doctoral student was utilized in the collection of the implementation relia-bility and inter-observer reliability data related to the implementation process of the sociometry and self-regulation scale. This student has taken part in many projects and has previously carried out the sociometry process with children. Before the implementation, the general structure of the self-regulation scale, the implementation reliability form, including the implementation steps, and the scoring process of the scale were explained to the doctoral student. Afterward, the consistency of the scoring was ensured through the sample video taken by the first researcher. Forms showing the sociometry implementation steps and a scoring chart were given for sociometry. The implementation reliability of the sociometry and self-regulation scale was found to be 100%.
This doctoral student participated in 100% of the sociometry implementation's pre-test, post-test, and follow-up data, and the inter-observer reliability was 100%. The same observer participated in 100% of the pre-test data, 70% of the post-test data, and 40% of the follow-up data of the self-regulation scale implementation. At this point, inter-observer reliability was found to be 100%. (p9, line 409-432).
Comment 15: In Social Validity heading (Place at end of findings. The Measure should be noted in the Measures section though. Where the interviews part of the pre-post measures or for social validity ???) They were also asked why the teacher interview for social validity was conducted only for children at risk of EBD.
Response 15: The social validity heading was placed at the end of the results and explained in detail. (p11, 525-544)
The social validity parent-teacher interview form developed for social validity was placed under the measures heading and explained in detail.
(2.4.5. Social Validity Parent-Teacher Interview Form
To obtain social validity data on the effectiveness of the intervention package, the researchers prepared the "Social Validity Parent and Teacher Interview Form." Through semi-structured interviews, the views of parents and teachers were obtained regarding the effects of the activities in the intervention package on children's self-regulation skills, problem behaviors, social skills, peer acceptance, and relationships with teachers. (p6, line 263-268)
Comment 16: In data analysis heading (How were the interviews analysed)
Response 16: (The data obtained from the interviews for social validity were analyzed with content analysis) (p10, line 437-438)
Comment 17: Results line 289 -355. See comments on text. Given the very small number of children participating, the pre-post test design appears inappropriate (ie combining five EBD together and 5 TD together). Given the measures, a different research design may have provided baseline and the effects of intervention more clearly for individual children and as a group. Overall though, I personally would have rather seen the children’s actual scores in Table format so I could compare across phases and also across children (both EBD and TD).
Response 17: The actual scores of all participants are given in Table 3 in the supplemental material file. The interpretation of this table is included in the results.
(The pre-test, post-test, and follow-up test scores of all participants on the dependent variables are shown in Table 3. Based on these scores, it was observed that there was an increase in self-regulation, social skills, and peer acceptance variables, as well as a de-crease in the problem behavior variable from pre-test to post-test for children at risk of EBD. Similarly, children with TD showed an increase in self-regulation, social skills, and peer acceptance variables and a decrease in the problem behavior variable between the pre-test and post-test stages. However, no change was observed in children who scored high in the pre-test stage in positive variables or who scored low in the pre-test stage in the problem behavior variable. Changes in the student-teacher relationship variable were generally low in both groups.
In general, the self-regulation scores of children at risk of EBD at the pre-test stage were lower than their peers with TD. At this point, it is seen that the second participant scored high in his group, while the eighth participant scored low. It can be said that the second participant scored high on the self-regulation scale because she was a student with internalized behavioral problems but did not have any problems in terms of attention and impulse control. In contrast, the second participant was an average child regarding social skills and peer acceptance, but his self-regulation skills needed support. (p10, line 440-456)
Comment 18: In Discussion (Line 386 – there is a statement about the second participant – no other participants have been mentions so this feel out of place. As noted on text and in this summary, the children’s data/findings need to be more personalised so the mention of a specific participant is understood by the reader.)
Lines 402 -423. This paragraph has too many ideas running through it so it is confusing to read. Separate to make each point clear and relate to the findings and literature.
Response 18: The actual scores of all participants are given in Table 3. In addition to the explanation about the second participant, the comment about the eighth participant was added to the discussion to personalize the findings of the children. (Therefore, considering subject subjectivity, children who did not have a predominant problem behavior among TD children, such as Participant Eight, who performed averagely in social skills but had low self-regulation skills, may have been selected. (p12, line 572-574)
The paragraph with many ideas has been separated. (p 13, line 602)
Comment 19: Limitations (lines 425-440) were appropriately identified as is further studies (not different).
Response 19: The requested change was made. (p13, line 619, 626, 630)
Comment 20: It was requested to mention as a limitation that children with TD were selected based on the teacher's statement.
Response 20: (Another limitation of this study may be that TD children were selected based solely on teachers' opinions. In further studies, it is recommended that all dependent variables for these children be determined by setting various prerequisites and using various measurement tools.) (p14, line 631-634)
Comment 21: (A conclusion is missing and there needs to be one included.)
Response 21: A concise conclusion was written. (In summary, it is known that many comprehensive social-emotional intervention programs have positive results on problem behaviors. In addition, the relationship between self-regulation and positive social behaviors has been shown. Although this intervention package, which directly targets the development of self-regulation skills, is not comprehensive, it has significantly contributed to the literature by reducing the problem behaviors of children at risk for EBD and improving their social skills and peer acceptance. In addition, it is important that the intervention package consists of play-based activities that can be easily implemented in the classroom and guide preschool teachers in implementation.) (p14, line 635-644)
Reviewer 2 Report
Comments and Suggestions for Authors
- Line 9: The acronym EBD is not defined when first introduced. Ensure consistency in defining acronyms throughout the paper.
- Before stating the study’s effects on key variables, provide background information and justify their significance.
- Clarify the criteria used to determine which five children were at risk of EBD and how others were classified as typically developing.
- Abstract lacks statistical results, making it unclear how the authors justify their claims. Including key statistical findings would enhance transparency.
- Line 26: EBD should be redefined, as the main paper is separate from the abstract.
- The in-text citation format appears inconsistent with the journal’s guidelines. Verify whether numbers or full names should be used.
- Line 45: The term early period is vague—specify the timeframe being referenced.
- Provide a clearer explanation of the types of intervention programs referenced, including specific methodologies or frameworks.
- The literature review relies on outdated sources and lacks recent advancements in self-regulation interventions. Incorporating newer studies, such as doi: https://doi.org/10.1016/j.neuroimage.2024.120740 and doi: 10.1017/S0954579424000725, would enhance its relevance.
- The paper does not adequately discuss how its findings contribute to the existing body of literature—this should be explicitly addressed.
- Does the study sufficiently justify the use of a single-group pretest-posttest design, given its potential threats to internal validity?
- Are the criteria for identifying children at risk of EBD sufficiently robust, and how might subjective teacher assessments affect reliability?
- With only 10 participants, how generalizable are the findings? Does the study have enough statistical power to detect meaningful effects?
- The research gap and study significance are not well articulated. Strengthening this section would clarify the necessity of the study.
- The scale used for measurement should be included as an appendix for reference.
- Tables summarizing validity and reliability results are missing and should be included to support the study’s methodological rigor.
Author Response
Thank you very much for your feedback, and we believe that all of the feedback has improved the research considerably.
Comment 1: (Line 9: The acronym EBD is not defined when first introduced. Ensure consistency in defining acronyms throughout the paper.)
Response 1: The requested change was made and all abbreviations in the text were checked. (Children at risk of emotional and behavioral disorders (EBD) p1, line 27-28
Comment 2: Before stating the study’s effects on key variables, provide background information and justify their significance.
Response 2: Effortful control was defined and its importance for children at risk of EBD was explained. (Effortful control is closely related to the concept of attention and emotion regulation. It is a structure that allows the individual to stop dominant reactions in favor of non-dominant but goal-related actions and to exercise conscious control over their behaviors (André et al., 2019). (p1, line 32-35)
For background, current literature on the social relationships and participation in learning activities of children at risk of EBD is provided. (Children at risk of EBD have low participation in learning activities and the ability to establish positive relationships with those around them, which affects the learning process (Hasty et al., 2023; Searle et al., 2014). (p1, line 38-40)
Comment 3: Clarify the criteria used to determine which five children were at risk of EBD and how others were classified as typically developing.
Response 3: All participants were selected primarily on the basis of teacher nomination. Then, selection was based on multiple data sources such as standardized scale, interview and observation for children at risk of EBD. This is explained in detail in the participants section. (The initial selection of children with both EBD and TD was based on the nominations of counselors and classroom teachers. Teachers were asked to nominate children in both groups if they did not have a disability diagnosis. However, if the child at risk of EBD exhibited internalized or externalized behavioral problems for at least three months, this situation negatively affected the child's educational performance and social relations. The fact that children with TD were not referred in this respect shows that these children were not at risk of EBD. (p4, line 141-147)
Comment 4: Abstract lacks statistical results, making it unclear how the authors justify their claims. Including key statistical findings would enhance transparency..
Response 4: Any requested changes were added to the summary section. (It was found that the intervention package was effective in self-regulation, problem behavior, social skills, and peer acceptance variables of children at risk of EBD (z=-2.02, p<.05) but not in student-teacher relationships (p>.05). In addition, in the post-test and follow-up tests, children at risk of EBD reached similar performance levels with TD children in terms of all variables (p>.05) is essential evidence showing the effectiveness and social validity of the intervention.) (p1, line 14-20)
Comment 5: Line 26: EBD should be redefined, as the main paper is separate from the abstract.
Response 5: The requested change was made. (Children at risk of emotional and behavioral disorders (EBD) (p1, line 27)
Comment 6: The in-text citation format appears inconsistent with the journal’s guidelines. Verify whether numbers or full names should be used.
Response 6: APA format is followed for in-text citation format. Studies with three or more authors are cited as first author et al.
Comment 7: Line 45: The term early period is vague—specify the timeframe being referenced.
Response 7: The term “early period” has been replaced with “preschool period”
Comment 8: Provide a clearer explanation of the types of intervention programs referenced, including specific methodologies or frameworks.
Comment 9: The literature review relies on outdated sources and lacks recent advancements in self-regulation interventions. Incorporating newer studies, such as doi: https://doi.org/10.1016/j.neuroimage.2024.120740 and doi: 10.1017/S0954579424000725, would enhance its relevance.
Response 8 and 9: A description of the general structure and characteristics of comprehensive programs has been added. (Although these programs are designed as social-emotional or school readiness intervention programs in terms of their general structure, they include various goals to support self-regulation skills. Such programs can intervene in the problem behaviors of children at risk while taking universal measures for children with typical development (Powell et al., 2006). In addition, these comprehensive intervention programs require teacher or parent training and include various intervention-specific materials. (p2, line 54-60)
In addition, research findings from two comprehensive preschool readiness programs are included to both explain comprehensive programs and to provide examples of more recent self-regulation interventions. (The literature shows that the content of comprehensive school readiness intervention programs includes early literacy skills, prosocial behaviors, and self-regulation skills. The Chicago School Readiness Project, which requires teacher training, examined the self-regulation and school readiness skills of 602 children in 35 Head Start classes over a year. As a result of the study, it was revealed that attention, inhibitory control, and ex-ecutive function skills were developed about self-regulation, and word, letter naming, and math skills were developed about school readiness (Raver et al., 2011). The Kids in Transition to School (KITS) intervention program, which includes both child and family training, includes early literacy skills, prosocial behaviors, and self-regulation skills such as coping with frustration, impulse and attention control and following instructions. In their study, Pears et al. (2012) implemented the KITS program for children with exter-nalizing behaviors. As a result of the research, it was found that there was an im-provement in the positive behaviors and self-regulation skills of the children in the experimental group who participated in the program compared to the children in the control group.) (p2, line 80-93)
Comment 10: The paper does not adequately discuss how its findings contribute to the existing body of literature—this should be explicitly addressed.
Response 10: A separate conclusion was created in order to more clearly demonstrate the contribution of this research to the literature. (In summary, it is known that many comprehensive social-emotional intervention programs have positive results on problem behaviors. In addition, the relationship between self-regulation and positive social behaviors has been shown. Although this intervention package, which directly targets the development of self-regulation skills, is not comprehensive, it has significantly contributed to the literature by reducing the problem behaviors of children at risk for EBD and improving their social skills and peer acceptance. In addition, it is important that the intervention package consists of play-based activities that can be easily implemented in the classroom and guide preschool teachers in implementation.) (p14, line 635-644)
Comment 11: Does the study sufficiently justify the use of a single-group pretest-posttest design, given its potential threats to internal validity?
Response 11: Thank you for pointing this out. We explained that we did not prefer a single-subject research design because the instruments used did not allow continuous data collection. We justified the appropriateness of the pretest-posttest model and explained the measures we took for internal validity.
(Since it was challenging to match children at risk for EBD in terms of behavioral problems, a single-group pretest-posttest model was preferred instead of a control-group pre-test-posttest model. In addition, since the data collection tools used in the study did not allow for continuous data collection, this model was decided upon rather than a sin-gle-subject experimental design. The literature states that this model is widely used to evaluate the effect of behavioral interventions (Cranmer, 2017). In this model, meas-urements regarding the dependent variables are taken from all participants before and after the intervention. The measurement results obtained are compared, and the sig-nificance of the difference between the results is examined. If there is a statistically sig-nificant difference between the results, it can be said that this difference is due to the intervention (Büyüköztürk et al., 2018).
Various measures were taken to control internal validity. The first researcher con-ducted the study with 10 participants who were continuing their education in the same school, so everyone was exposed to the same intervention conditions in the same envi-ronment. The intervention process lasted for two months. Thus, the maturation effect was controlled. In addition, the teachers were not informed about the details of the inter-vention before and during the intervention and were asked to continue their usual ed-ucation processes. Thus, other variables that may impact the intervention results were controlled. In addition, all test conditions were carried out in the same environment, with the same materials, and by the same person, as planned, with high implementation reliability. Thus, problems arising from test conditions, instruments, and scoring were prevented. (p.4-5, line 185-205).
Comment 12: Are the criteria for identifying children at risk of EBD sufficiently robust, and how might subjective teacher assessments affect reliability?
Response 12: In identifying children at risk for EBD, we did not rely solely on the teacher's nomination. We applied the SCBE-30 standard scale to the candidates nominated by the teacher as at risk. We conducted a functional assessment interview with the parent and teacher of the child at risk. From the parent, we obtained information about the characteristics and duration of the child's problem behaviors and that these behaviors damaged the child's social relationships within the family. From the teacher, after this interview, we obtained information that the behaviors exhibited by the child damaged his/her educational performance and peer relationships. After these, the first researcher attended the classroom of each child at risk for EBD for one day and observed the occurrence of problem behaviors in the classroom. After all these processes, we decided that five children were at risk of EBD.
Comment 13: With only 10 participants, how generalizable are the findings? Does the study have enough statistical power to detect meaningful effects?
Response 13: Thank you for your comment. We accepted the fact that the study had 10 participants as a limitation and suggested that further research should be conducted with larger groups.
(The limited number of participants is seen as the most critical limitation of this study. The results obtained due to the limited number of participants should be cautiously approached. It is essential for the validity of the results to apply the intervention package to a larger group of participants in further studies.) (p13, line 616-619)
Due to the limited number of participants, power analysis will not be possible.
However, in the wilcoxon and mann whitney u analyzes used, the expressions of effectiveness can be mentioned in the expression of the findings. In order not to generalize this and to indicate that it is limited to the findings of this study, the expression based on these findings is used. (line 439-524)
Comment 14: The research gap and study significance are not well articulated. Strengthening this section would clarify the necessity of the study.
Response 14: The importance and necessity of this research is explained with up-to-date references. (Therefore, it is thought that there is a need for classroom-based interventions in the literature. Therefore, there is a need to examine the effects of interventions that directly target only improving self-regulation skills.
It is stated in the literature that there are a limited number of self-regulation in-tervention studies in the preschool period (Baron et al., 2017; Morawska et al., 2019; Pandey et al., 2018). Although self-regulation skills are critical for all individuals throughout life, they are also important for students at risk of EBD. It is expected that by supporting the self-regulation skills of children at risk of EBD, their positive social behaviors and participation in ac-ademic activities will increase and, accordingly, their problem behaviors will decrease, their social acceptance by their peers will increase, and their conflicts or problems with their teachers will decrease (Edossa et al., 2018; Hasty et al., 2023; O’Donnell et al., 2024; Pandey et al., 2018; Salerni & Messetti, 2025). Considering that if the problem behaviors exhibited by children at risk of EBD are not intervened in the preschool period, the behaviors exhibited will continue to increase (Meagher et al., 2009; Reef et al., 2011), it is seen that these children need interventions that support their self-regulation skills. (p3, line 94-119)
Comment 15: The scale used for measurement should be included as an appendix for reference.
Response 15: Permission to use the measurement tools has been obtained, but permission to share them has not been obtained.
Comment 16: Tables summarizing validity and reliability results are missing and should be included to support the study’s methodological rigor.
Response 16: All the findings presented by the analysis program used in line with the analysis methods selected in accordance with the research questions are presented in full.
Round 2
Reviewer 2 Report
Comments and Suggestions for Authors
The authors have not adequately addressed the previous review comments. While minor modifications were made, such as adding three words in the abstract, serious efforts not carried out. The literature review contains irrelevant additions while still lacking the recommended suggestions. The justification for key methodological choices remains weak, and crucial clarifications requested in the initial review were either ignored or insufficiently addressed. The intervention's effectiveness is still not rigorously supported by statistical evidence, and the discussion lacks depth in contextualizing the findings within the broader literature. The authors must substantially revise the manuscript by incorporating the reviewer’s prior recommendations rather than making superficial changes. Without these improvements, the paper does not meet the necessary standards for publication.
Author Response
Thank you very much for your comments in both rounds. We have carefully reviewed your comments and tried to understand them using the literature. We have highlighted our corrections in green in the second round.
1- Regarding your comment about statistical additions regarding the findings in the abstract section, we made additions, considering that the journal's abstract length should not exceed 200 words (lines 17-20).
In the follow-up, no significant change was observed in all variables except problem behaviors. However, the levels were maintained (p>.05). Only problem behaviors variable showed a significant decrease compared to the post-test (z=-2.03, p<.05, r=0.64).
2- In the first round, we made several additions to the introduction to provide background to the research and reinforce its importance, but these were found to be insufficient. Therefore, in this round, we primarily explained the development of self-regulation and the factors related to developing these skills (lines 30-39).
Although self-regulation skills begin to develop at birth, they are based on external regulation during infancy. In this process, when infants give clues that they need regulation, their parents or primary caregivers regulate their needs. As children grow and mature, there is a shift from external to internal regulation. Factors such as children's maturation, the development of the prefrontal cortex and its activation of other regions of the brain, the secure relationship children have with their caregivers, the presence of peers and adults who model self-regulation skills in their environment, and the opportunities children have to practice self-regulation skills play an important role in this change process (Bernier et al., 2010; Florez, 2011; McClelland & Tominey, 2015).
3- Two sources were recommended in the first round, which were also carefully reviewed at the time but were not included because they were descriptive rather than intervention studies. In this round, key findings from these two sources are presented in the introduction to strengthen the conceptual framework (lines 40-56).
The development of the dynamic structure of the brain's functional networks, which are closely related to self-regulation, undergoes significant changes in early childhood, and these changes directly affect the brain's information processing capacity. In Song et al.'s (2024) study, significant age-related increases were observed in the temporal flexibility values of functional networks such as executive control networks, salience networks, and default mode networks related to self-regulation. This indicates that the networks gain more flexible structures over time, and the capacity to reorganize functional connections increases, reflecting the development of self-regulatory functions such as attentional control and cognitive flexibility. The increase in the spatiotemporal diversity of the salience network with age is associated with self-regulatory skills such as emotional awareness, switching between emotional states, and contextualizing behaviors. Childhood abuse is one of the important situations that affect the functional connectivity between the brain networks. Cao et al. (2024) found that abuse may affect the functional connectivity patterns in networks related to attention and emotion regulation in the long term. However, positive parenting may be a preventive factor. In conclusion, the findings from these studies suggest that dynamic changes in the brain's functional networks may form the neurological basis of self-regulation.
4- The rationale for choosing a single-group pretest-posttest design was explained in the first round. The rationale for not selecting a control group model or a single-subject experimental design as an alternative to this model was also explained. However, these explanations were found insufficient. In addition to these explanations, justification was provided for selecting this model by adding the information that access to children with EBD risk in Türkiye is difficult (lines 215-218).
In Türkiye, children with EBD cannot officially benefit from special education services. In addition, children at risk of EBD are not systematically screened and monitored in schools. Therefore, the difficulty in identifying children at risk of EBD is an important factor in selecting this model.
5- We also know and accept that this model is referred to as a weak experimental design in the literature. Therefore, we avoided generalizations and assertive statements. The results section does not contain assertive statements and is formed only by expressing the data in the tables reflecting the nonparametric test results.
6- Based on your comment in the first round, “Tables summarizing validity and reliability results are missing and should be included to support the study's methodological rigor.” and your comment in the second round, “The intervention's effectiveness is still not rigorously supported by statistical evidence,” two primary sources were carefully reviewed, but no information was found regarding validity and reliability tables in nonparametric tests (Fraenkel, Wallen, & Hyun, 2012; Gravetter & Wallnau, 2013). If there is a resource you can recommend on this subject, we are ready to review it and make arrangements accordingly.
7- In the data analysis section, we explained that we used nonparametric tests because we could not meet the assumptions of parametric tests (lines 468-472).
Parametric tests require assumptions about specific population parameters, such as normal distribution, and test hypotheses about these parameters. In cases where these assumptions are not met, various hypothesis testing techniques called nonparametric tests are used as alternatives to parametric tests. This study used nonparametric statistics to consider the number of participants (Gravetter & Wallnau, 2013).
8- In the Discussion section, a conclusion sentence emphasizing the limitations was written so that the first sentence would not be perceived as assertive and was discussed with the literature in line with your comment (lines 583-591).
Within the study's limitations, the intervention may be effective in the self-regulation, social skills, peer acceptance, and problem behaviors of children at risk for EBD. However, it is not effective in their relationships with their teachers. The content of the intervention package included objectives such as focusing, maintaining and controlling attention, inhibitory control, controlling negative emotions, or acting by the instructions and rules, which are various, comprehensive, and compatible with the intervention contents in the literature, maybe a variable that explains the overlap of the effectiveness findings of this intervention with the literature (Barnett et al., 2008; Domitrovich et al., 2007; Pears et al., 2012; Raver et al., 2011).
9- Based on your comments, we discussed our findings within the broader literature in the discussion section (lines 621-639, lines 644-647).
The activities carried out for the objectives were play-based, which may have increased children's self-regulation performance. Play is a valuable tool that provides the appropriate context for supporting self-regulation skills and allows children to control their mental activities (Bodrova et al., 2013; Vallotton & Ayoub, 2011). The implemented intervention package consisted of fun and collaborative activities may have supported children's ability to express their ideas, explain their reasoning processes, and talk about their learning processes. These important skills may have improved children's self-regulation (Pino-Pasternak et al., 2014).
Features such as the practitioner's approach, how she presented the activities, and the language of communication she used with the children may also have increased children's self-regulation performance. In the process, the practitioner provided children with opportunities to explain their thoughts and thinking processes, modeled thinking processes or strategies by thinking aloud, gave children opportunities to practice, and provided feedback. In the literature, it is stated that all practices, such as adults giving children the opportunity to make choices and express their ideas clearly (Chan et al., 2014; Green et al., 2011), modeling related processes by thinking aloud (Florez, 2011; Wery & Nietfeld, 2010), giving children opportunities to practice, providing support at the level needed, reinforcing successful attempts, and gradually withdrawing the support provided all support self-regulation skills (Florez, 2011; Rosanbalm & Murray, 2017).
Improvement in self-regulation skills may have been effective in reducing children's behavioral problems. Many studies show that self-regulation skills can effectively prevent and reduce behavioral problems (Bierman et al., 2008; McClelland et al., 2017; Reyes et al., 2012).
Round 3
Reviewer 2 Report
Comments and Suggestions for Authors
The authors have made considerable improvements, successfully addressing the previous concerns. I am happy to recommend this revised version for publication and commend the authors for their excellent work.